# Analysis of Givinostat/ITF2357 Treatment in a Rat Model of Neonatal Hypoxic-Ischemic Brain Damage

**DOI:** 10.3390/ijms23158287

**Published:** 2022-07-27

**Authors:** Paulina Pawelec, Joanna Sypecka, Teresa Zalewska, Malgorzata Ziemka-Nalecz

**Affiliations:** Mossakowski Medical Research Institute, Polish Academy of Sciences, 5, A. Pawinskiego Str., 02-106 Warsaw, Poland; ppawelec@imdik.pan.pl (P.P.); jsypecka@imdik.pan.pl (J.S.)

**Keywords:** neonatal hypoxia-ischemia, HDAC, histone deacetylase inhibitor, givinostat/ITF2357, microglia, fractalkine/CX3CL1, cytokines, inflammation

## Abstract

The histone deacetylase inhibitor (HDACi) Givinostat/ITF2357 provides neuroprotection in adult models of brain injury; however, its action after neonatal hypoxia-ischemia (HI) is still undefined. The aim of our study was to test the hypothesis that the mechanism of Givinostat is associated with the alleviation of inflammation. For this purpose, we analyzed the microglial response and the effect on molecular mediators (chemokines/cytokines) that are crucial for inducing cerebral damage after neonatal hypoxia-ischemia. Seven-day-old rat pups were subjected to unilateral carotid artery ligation followed by 60 min of hypoxia (7.6% O_2_). Givinostat (10 mg/kg b/w) was administered in a 5-day regimen. The effects of Givinostat on HI-induced inflammation (cytokine, chemokine and microglial activation and polarization) were assessed with a Luminex assay, immunohistochemistry and Western blot. Givinostat treatment did not modulate the microglial response specific for HI injury. After Givinostat administration, the investigated chemokines and cytokines remained at the level induced by HI. The only immunosuppressive effect of Givinostat may be associated with the decrease in MIP-1α. Neonatal hypoxia-ischemia produces an inflammatory response by activating the proinflammatory M1 phenotype of microglia, disrupting the microglia–neuron (CX3CL1/CX3CR1) axis and elevating numerous proinflammatory cytokines/chemokines. Givinostat/ITF2357 did not prevent an inflammatory reaction after HI.

## 1. Introduction

Hypoxic-ischemic encephalopathy (HIE) in newborn infants is one of the most important causes of death and/or long-term neurologic sequela, such as cerebral palsy and neurobehavioral and cognitive dysfunction. The initiation and development of neonatal brain damage following HI is a complex process with multiple contributing mechanisms and pathways resulting in both early and delayed injury. The role of both acute and chronic inflammation as potential drivers of tissue damage has been recognized in the pathogenesis of HIE. Inflammation is driven primarily by the activation of inflammatory glial cells residing in the central nervous system (CNS) together with infiltrating cells of the peripheral immune system (mast cells, monocytes/macrophages) to produce several proinflammatory factors, such as cytokines, chemokines, reactive oxygen and nitrogen species, which are pivotal mediators of persistent neuronal injury [1,2]. The pathophysiological role of inflammation has been supported by some clinical studies showing an increase in cytokine and chemokine serum levels in newborns with ischemic brain injury. Despite the fact that numerous in vivo and in vitro studies provide growing evidence that anti-inflammatory compounds show promising neuroprotective effects in animal models, only a few of them have progressed to clinical trials. The application of potential neuroprotective agents has been truly restricted due to insufficiency and/or serious side effects demonstrated by their impact on normal brain activity. However, even though animal models are poorly predictive of human outcomes, these experiments are mandatory, especially in drug development. The only available effective treatment, therapeutic hypothermia, neither provides complete brain protection nor stimulates the repair necessary for neurodevelopmental outcome. Despite hypothermia treatment, about 50% of neonates with moderate or severe hypoxia-ischemia experience disability or death; thus, therapies that improve outcomes are extremely needed. In recent years, several strategies for the treatment of neonatal HIE were tested in clinical trials (e.g., erythropoietin, allopurinol, melatonin, cannabidiol, doxycycline, minocycline, exendin-4/exenatide); however, most of them cause side effects (e.g., arthralgia, embolism and thrombosis, hypertension, influenza-like illness, skin reactions, abnormal behavior, insomnia, fever, diarrhea, vomiting, tremor, stroke) (Victor et al., 2022). Therefore, there is still an urgent need to identify new compounds that may be hopefully adapted as a therapeutic option in infants with hypoxic-ischemic insult [3,4]. Consistently, a number of studies have pointed to broad-acting inhibitors of histone deacetylases as anti-inflammatory agents. Thus, several histone deacetylase inhibitors (HDACis) were tested and found to have neuroprotective actions in several rodent models of acute and neurodegenerative diseases involving the ischemic adult brain [5,6,7,8]. A few available reports have addressed the potential use of this line of treatment in hypoxia-ischemia-injured immature brains [9,10]. The neuroprotective action of HDAC inhibitors is multifold and involves the modulation of oxidative pathways and components of excitotoxicity and a reduction in several inflammatory and proapoptotic factors. The aim of the present study was to elucidate whether a new compound, a histone deacetylase inhibitor, Givinostat (also known as ITF 2357), can diminish neural inflammatory responses in vivo in neonatal HI together with the molecular consequences of this action. We focused on this compound due to its favorable safety profile in humans and its use in phase II clinical trials in children with active systemic onset juvenile idiopathic arthritis [11].

## 2. Results

### 2.1. The Effect of Givinostat on IL-1β Expression in Primary Glial Cells Exposed to LPS

First, we assessed the response of the inflammatory cytokine IL-1β to the presence of Givinostat. The expression of IL-1β was induced by LPS in primary glial cell culture. The results illustrated in Figure 1 and a magnified photograph show that the presence of 1 µM or 10 µM Givinostat in incubation medium reduced the LPS-induced expression of this cytokine to the control level. Simultaneously, Givinostat had no influence on the basal expression of IL-1β. Therefore, we hypothesized that there is a strong link between inflammatory processes and Givinostat action.

### 2.2. The Effect of Givinostat on the Acetylation of Histone H3

The initial in vitro findings were further extended to in vivo using a well-established model of neonatal HI in rats. To confirm the efficiency of Givinostat in inhibiting histone deacetylase, we determined the level of acetylated histone H3 (AcH3) within the ipsilateral (hypoxic-ischemic) and control hemispheres. Figure 2 depicts a densitometric analysis of representative immunoblots probed with an antibody specific to acetylated histone H3. As shown in the respective Figure 2, 24 and 72 h following hypoxia-ischemia combined with or without Givinostat, AcH3 immunoreactivity levels did not differ from the control level. A noticeable reduction in acetylated histone H3 to approximately 50% was observed 5 days after injury alone (*p* < 0.01, ctr vs. HI), whereas in animals treated with Givinostat, the value for AcH3 reached the control level (*p* < 0.01, HI vs. HI+Giv).

### 2.3. The Effect of Hypoxia-Ischemia and Givinostat Treatment on the Activation and Polarization of Microglia

We further explored the effect of Givinostat administration on the activation of microglia/macrophages after HI. For this purpose, we performed ED1 staining on coronal brain sections of the control brain (sham operated), HI and HI combined with Givinostat. The data presented in Figure 3 show numerous ED1-positive cells in the ipsilateral damaged hemisphere at 7 and 14 days after HI. Most microglial cells were ameboid in shape with thick processes and were considered to be in an activated state. The activated microglial cells were scattered throughout the entire cortex and corpus callosum. In contrast, in slices obtained from control brains, we observed only single ED1-positive cells. The same activation pattern was demonstrated after Givinostat treatment.

Depending on the surrounding microenvironmental signals, microglia can acquire distinct phenotypes (microglial polarization), presenting diverse activation states and different modes of action. Therefore, we next examined whether Givinostat promotes the polarization of microglia from the M1 to M2-like phenotype at 7 and 14 days after HI. For this purpose, we performed double staining with an IL-1β antibody coupled with ED1 to identify the activated proinflammatory M1 phenotype and ED1/arginase-1 to identify the anti-inflammatory M2-like phenotype. Seven and fourteen days after HI, approximately 50% of the ED1-positive cells expressed IL-1β in the ipsilateral hemisphere (Figure 4), and 30% of the microglial cells were stained positively with ED1/Arg-1 (Figure 5). The administration of Givinostat after HI did not change the number of ED1/IL-1β and ED1/Arg-1 cells in the ipsilateral hemisphere. This finding indicates that Givinostat had no influence on the microglial polarization state after neonatal HI.

### 2.4. The Effect of Hypoxia-Ischemia and Givinostat Treatment on Microglia–Neuron Interactions

The binding of neuronal fractalkine (CX3CL1) to its microglial receptor (CX3CR1) is responsible for the interaction between neurons and microglial cells, the process engaged in the regulation of microglial activation. Thus, we next estimated the effect of HI and Givinostat administration on the colocalization of the receptor CX3CR1 with its ligand, fractalkine. To address this issue, we performed an immunohistochemical study. As shown in the respective Figure 6, a detailed confocal photomicrograph analysis of the control animals showed that 80% of the cells expressing fractalkine colocalized with its receptor CX3CR1, suggesting that under normal conditions, both proteins remain in close contact. At 7 and 14 days after HI injury, the number of cells presenting both factors in the ipsilateral hemisphere was reduced to 50 and 55%, respectively, compared to the control. Givinostat administration did not influence the number of double-stained CX3CL1/CX3CR1 cells at 7 days or 14 days after insult (Figure 6). The reduced colocalization of the receptor with its specific ligand may imply the impairment of contact between neurons and glia after neonatal HI.

Next, we assessed whether the diminished colocalization was due to the decreased level of CX3CR1 in microglia using antibodies against Iba1 (a marker for all types of microglial cells, resting and activated) or ED1 (a marker for activated, macrophage-like cells) and an antibody against CX3CR1. As shown in Figure 7, in control animals, approximately 50% of the Iba-positive microglial cells coexpressed CX3CR1. HI resulted in a time-dependent decrease in double-stained cells. A decreased number of Iba1/CX3CR1-positive cells was observed at 7 days after HI. The prolongation of animal survival until 14 days after injury strengthened this reductive effect and decreased the number of immunopositive Iba1/CX3CR1 cells to 35% compared to the control. A confocal examination of the control and damaged hemispheres clearly showed that Givinostat did not change the number of immunoreactive Iba1(+)/CX3CR1(+) cells. Similarly, a lack of reaction to Givinostat combined with HI was also observed in the activated microglia stained with the ED1 marker and CX3CR1 (Figure 8).

### 2.5. The Effect of Givinostat on the Levels of Cytokines and Chemokines in Rat Brains after HI

Finally, we analyzed the levels of several proinflammatory and anti-inflammatory cytokines and chemokines (as enumerated in the Materials and Methods section) by Luminex and Western blot methods. The potential modulation of these factors either by HI or HI combined with Givinostat was investigated over time: at 24 h, 72 h and 5 days after the insult in the damaged cerebral hemispheres and compared to the age-matching controls. The results of the Luminex analysis are shown in Figure 9 (proinflammatory cytokines), Figure 10 (anti-inflammatory cytokines) and Figure 11 (chemokines). The analysis of the fractalkine expression (WB assay) is shown in Figure 12. In general, we observed individual variations in cytokine levels, and not all responded to insult induction at the investigated time points. A significantly decreased amount of IL-1β at 24 h was noted in the damaged hemisphere (*** *p* < 0.001 HI vs. control) (Figure 9). In contrast, simultaneously, the chemokines MIP-1α and MCP-1 were upregulated (*p* < 0.01 and *p* < 0.0001, respectively) compared to the control value (Figure 11). At this time point, the expression patterns of the other analyzed inflammatory and anti-inflammatory mediators did not differ from those of the control. Notably, several cytokines and chemokines were elevated at 72 h after injury but with different degrees of significance compared to the age-matched controls. A clear increase in MIP-1α protein expression was observed. The level of this chemokine was approximately 10-fold higher than the control value at 72 h after HI (*p* < 0.0001) (Figure 11). At the same time point, we also observed increased levels of the proinflammatory cytokines IL-1α (*p* < 0.05), IL-1β (*p* < 0.05), IL-5 (*p* < 0.001), IL-12 (0.05) and INFγ (*p* < 0.05); the anti-inflammatory cytokines IL-4 (*p* < 0.05) and IL-13 (*p* < 0.05); and the chemokines MCP1 (*p* < 0.05) and CX3CL1 (WB assay) (*p* < 0.01) compared to the control values (Figure 9, Figure 10, Figure 11 and Figure 12). The amount of nearly all the investigated factors at 5 days after insult generally presented the age-matched control level, with the exception of the higher expression of CX3CL1 (*p* < 0.05). The levels of IL-2, IL-7, IL-18, TNF-α, CXCL-1, G-CSF, M-CSF, CCL-5, MIP-3α and VEGF did not change after HI at any investigated time point (results not shown). Givinostat combined with hypoxia-ischemia did not further change the amount of investigated pro- and anti-inflammatory cytokines. The only effect of the HDACis was observed in the case of MIP-1α. The level of this chemokine was significantly suppressed to approximately 50% after Givinostat administration (*p* < 0.001, HI vs. HI+Giv) at 72 h after HI (Figure 11).

## 3. Discussion

In the present study, we showed that administration of the histone deacetylase inhibitor Givinostat did not diminish the extent of inflammatory responses induced after neonatal HI. The lack of effect remains in contrast with convincing evidence that HDAC inhibitors are generally efficacious neuroprotective agents in adults and neonatal brain injury associated with inflammation [5,8,12,13]. In addition, a study focused on determining Givinostat function described its ability as a potent anti-inflammatory agent in addition to its protective effect in in vitro and in vivo experimental models of different injuries [14,15,16,17,18].

At the initial step of our study, we determined the capacity of Givinostat to inhibit the release of inflammatory cytokines in glial cell cultures treated with LPS. Our results showed a positive effect of 1 and 10 µM Givinostat that involved a substantial decrease in IL-1β expression potentiated by LPS. This event may reflect an anti-inflammatory action of this agent. Notably, there are also interesting data indicating that some inhibitors of histone deacetylases, e.g., TSA, SAHA and M344, demonstrated, in contrast to our results, a potentiation of the LPS-stimulated inflammatory response [19]. These findings illustrate the complex consequences depending on the experimental conditions.

To determine whether the anti-inflammatory action of Givinostat demonstrated in our in vitro experiment may lead to reduced inflammation and thus may promote neuroprotection in vivo, we examined the effect of this agent in a model of neonatal HI. First, we examined the effect of HI and Givinostat treatment on the level of histone H3 acetylation. We observed a positive effect of this HDACi through the increased acetylation of histone H3, which diminished after hypoxic-ischemic injury. Our principal findings showed that the Givinostat treatment of neonatal HI did not decrease the number of ED1-positive cells (microglia/macrophages) in the damaged ipsilateral hemisphere at 7 and 14 days after insult. The majority of microglial cells presented morphological features characteristic of activated microglia. Active microglial cells migrate to the site of injury and act in concert with recruited monocytes/macrophages to elicit an immune response. Both cell types play an important role in the proinflammatory cytotoxic reaction, which may exacerbate the ischemia-induced injurious effect after HI [1,10,20,21]. Of note, only single ED1 positive cells were detected in the control brain.

Contrary to the general notion of microglial toxicity in the neonatal brain following HI, microglia also produce beneficial effects, indicating highly complex microglial functions. Inflammatory signals are an important innate mechanism responsible for the elimination of pathogens and debris, the remodeling of synapses and assisting in tissue repair by maintaining neurogenesis and organized brain circuits during the postnatal periods [22,23,24,25,26,27,28]. Therefore, the complete suppression of microglial activity might lead to more extensive brain damage after ischemic injury [29,30]. 

The complexity of microglial function may contribute to varying sensitivities to microenvironmental signals. In the neonatal brain, microglial cells undergo a morphologic transition after HI injury from the proinflammatory M1 to the alternative M2 phenotype, leading to neurotrophic and anti-inflammatory signaling, which may attenuate brain injury [13,31,32,33]. As demonstrated in the current study, at 7 and 14 days after HI, approximately 50% of the cells presented a positive reaction with the ED1/IL-1β marker characteristic for the M1 phenotype, whereas 30% of the cells were labeled with ED1/arginase-1, indicating phenotype M2. The treatment of rats with Givinostat does not facilitate the further conversion of acquired phenotypes and therefore is not attributed to the suppression of toxic microglial activation. This fact remains in contrast with previous reports, in which the treatment of neonatal HI with different HDAC inhibitors, sodium butyrate (SB) or scriptaid, promoted the polarization of microglia into the M2 phenotype and inhibited inflammatory responses [12,13,34]. The contradictory results described above may be due to the inhibitor-specific action associated with upregulation of the pattern of genes related to pro- or anti-inflammatory effects. Thus, microglia display a wide spectrum of activation states, and according to the statement of some researchers, they are oversimplified by using the M1/M2 classification [35,36,37]. 

A large body of data consistently shows a strong link between inflammatory processes and the neural fractalkine (CX3CL1) signaling pathway mediated by its unique receptor CX3CR1, present predominantly on microglial cells. The reciprocal interaction between these two factors allows precise and effective communication between neurons and microglial cells and thus plays a key role in coordinating many aspects of brain functions, e.g., influencing synaptic maturation during development and plasticity and controlling immune processes [38,39,40]. Recently, fractalkine and its receptor (CX3CR1) signaling were shown to function as endogenous inhibitory signaling pathways that can maintain microglia in the quiescent “off” state and thus inhibit the release of inflammatory cytokines [41,42,43,44]. Moreover, treating microglial or mixed glial cultures with the soluble fractalkine isoform suppresses the LPS-induced activation of microglial cells and reduces the production of inflammatory factors [45,46,47]. However, several independent studies show that the interruption of the CX3CL1/CX3CR1 signaling pathway in ischemic adults brought contradictory results—neuroprotective or detrimental effects—due to different experimental conditions [48,49,50,51]. Furthermore, there are also data postulating that the disruption of CX3CL1/CX3CR1 communication by the deletion of the cx3cr1 gene causes neurotoxicity in mouse models of systemic inflammation PD and ALS [52] but protects against neuronal loss in a mouse model of focal cerebral ischemia [51]. The current study, performed on immature rats, illustrates a clear colocalization of the interaction between the CX3CR1 microglial receptor and its ligand, CX3CL1, in control animals. As seen on the immunohistochemical images, HI led to the loss of colocalization between the two investigated factors. In addition, treatment with Givinostat did not alter the interruption of the HI CX3CL1/CX3CR1 signaling pathway. The reduced interaction did not allow us to identify their protective or detrimental effects. However, based on the high number of activated M1 microglia, we suspect that the loss of interaction may produce detrimental effects in neonatal HI. The specific response most likely depends on the local concentration of CX3CL1 and CX3CR1 [40]. Moreover, genetic ablation of the fractalkine receptor showed increased microglial activation in different models of inflammation [52,53].

We found an increased level of brain fractalkine after neonatal HI; however, most of the available data reported that the function of fractalkine is focused mainly on physiological conditions in the brain. The isoforms of fractalkine may exhibit distinctive biological activities associated with specific mediators. Fractalkine seems to contribute to the remodeling of neuronal circuits and the maturation of excitatory glutamatergic synapses during postnatal development [23,54,55]. Many studies suggest that CX3CL1 has dual activities in the CNS with either beneficial or detrimental potentials depending on the activation states of the microglia. Lauro et al. [56] demonstrated that CX3CL1 drives microglia towards an anti-inflammatory phenotype, reducing the expression of proinflammatory genes and increasing the expression of genes related to the anti-inflammatory state. Fractalkine also limits neuronal damage, counteracting excitotoxicity [57,58]. Therefore, the release of fractalkine may be a physiological response of brain tissue to trigger neuroprotection. However, despite the increased level of fractalkine, there is still no sufficient information regarding its contribution to inflammation in neonatal HI.

In addition to fractalkine, numerous other chemokines are produced and released by neurons and glial cells in the immature CNS. Only a few data are available on their potential role in the development of HI. We focused particular attention on the expression of two members of the beta family chemokines—MCP-1 (monocyte chemoattractant protein-1), also known as CCL2, and MIP-1α (macrophage inflammatory protein-1 alpha). Both chemokines have been implicated as potential modulators of the inflammatory response associated with the recruitment of monocytes in the neonatal brain [59,60]. 

A marked increase in both chemokine levels in the tissue obtained from the lesioned hemispheres during our study remains in general agreement with findings reported by other studies [59,61,62]. However, the temporal patterns of their stimulated expression after neonatal brain injury showed some differences. In our study, the peak of MCP-1 immunoreactivity was detected at 24 h after injury, whereas MIP-1α, barely detectable in the control, presented the highest level at 72 h postinjury. Thereafter, the immunoreactivity declined to the control level. However, another study revealed stimulation of MIP-1α up to 5 days after ischemic insult [62]. In addition, there is definitive evidence that increased MCP-1 in the adult brain exacerbates ischemic injury and is associated with recruitment of inflammatory cells [63,64,65]. The most important finding of our investigation is a clear reduction in the postischemic level of MIP-1α after Givinostat administration, most likely indicating some immunosuppressive effect. The same procedure did not change the amount of MCP-1. Given its well-documented proinflammatory role, MCP-1 seems to be an unlikely candidate for inducing neuroprotection. The pathophysiological inflammatory role of MCP-1 in neonatal brain injury is evident from protection by functional inactivation of MCP-1 after insult [66,67] or in mice with depleted IL-1 converting enzyme [61]. The different effects of the HDACis on the investigated chemokines in the damaged neonatal hemisphere are difficult to explain, which may be due to the different mechanisms of regulation in different cell types [68]. This prediction may be reinforced by data reported by Cowell et al. [62], who showed that MIP-1α was restricted to cells of the monocyte/macrophage lineage, whereas MCP-1 was detected in multiple cell types, including neurons. Another set of informative data came from examining valproic acid-treated adult rats in intracerebral hemorrhage. The authors found downregulation of mRNAs of both chemokines, MIP-1α and MCP-1 [69]. This finding supports the concept that the effect of HDACi application depends on multiple models of injury and the specificity of the inhibitor.

The response of microglial activation after HI injury is accompanied by an increase in cytokines in the ipsilateral hemisphere. Cytokines work as a final common pathway to injury and are involved in numerous functions involving homeostasis, regulating immune cell proliferation and differentiation and adjusting inflammatory cell function [70]. 

In the current study, we focused on the temporal expression profile of proinflammatory and anti-inflammatory cytokines. We found that the expression of only a few of them increased significantly, mostly pronounced at 72 h post-insult. Among the proinflammatory cytokines, we found upregulation of IL-1α, IL-1β, IL-5, IL-6, IL-12, IL-17 and IFNγ. The unchanged level of some estimated proinflammatory cytokines did not exhibit changes and did not differ from the controls and likely did not play a prominent damaging role in our experimental conditions. Our results are in general accordance with other reports, depicting a considerable alteration in cytokine expression after neonatal hypoxia-ischemia [1,60]. This observation emphasizes that cytokines could be important mediators of HI-related brain injury. There is an exception in IL-6, which appears to be a proinflammatory mediator or, depending on the context, downregulates inflammation [70]. Notably, in our study, at 24 h after HI, only IL-1β was downregulated. This downregulation might imply differences in its level to the decreased production by different cell types (e.g., activated glial cells, neurons and microvascular endothelial cells) and infiltration of circulating immune cells into the brain parenchyma [1]. However, our estimation method does not provide information regarding the anatomical and cellular distribution of IL-1β. The early reduction in IL-1β may be particularly important and may play a neuroprotective function in the development of encephalopathy and may counteract the initial inflammatory response of other cytokines.

A number of reports point to the damaging role of TNFα activated by brain ischemia. In neonatal rats with HIE, the levels of TNFα in the serum and lesioned brain are significantly increased as early as 4 h after injury [71,72]. Hence, TNFα may play a detrimental role in mediating the initial inflammatory response in neonatal HI. Our first assay was performed at 24 h, and we did not detect changes in the level of TNFα until 5 days.

The neuroimmune response is a complex balance between pro- and anti-inflammatory cytokines. Cytokines that antagonize inflammatory responses are IL-4, IL-10 and IL-13. We found an increase in the IL-4 and IL-13 levels at 72 h postinjury, the time when the selected proinflammatory cytokines were elevated. The increase in IL-4 and IL-13 may function as a global endogenous defense but is insufficient to reduce inflammation after ischemia. Based on our findings, the treatment of HI neonatal rats with Givinostat does not change the amount of either pro- or anti-inflammatory cytokines.

In contrast to our findings, the inhibition of histone deacetylase ITF3056, specific for HDAC8 or ITF2357/Givinostat, reduced proinflammatory cytokine production and systemic inflammation in other in vitro and in vivo experiments [14,73].

## 4. Materials and Methods

### 4.1. Primary Glial Cell Culture (In Vitro Experiments)

Primary mixed glial cultures were prepared from postnatal day 1 (PND1) rats. Isolated brains were washed with a cold phosphate-buffered saline buffer (PBS) and were then dispersed by pipetting in cold DMEM GlutaMAX High Glucose (Gibco) with the addition of 10% fetal bovine serum (FBS; Gibco) and 1% antibiotic-antimycotic solution (AAS; amphotericin B, penicillin, streptomycin; Gibco). The mechanically fragmented tissue was filtered (40 μm Hydrophilic Nylon Net Filter) and the obtained cell suspension was seeded into 6-well plates previously coated with poly-L-lysine (PLL). The culture was maintained for 10-11 days in a constant humidity incubator (21% O_2_, 2.5% CO_2_, 37 °C). Afterwards, bacterial lipopolysaccharide (LPS) was added to the culture medium at a concentration of 0.3 µg/mL. Simultaneously, the cells were incubated with Givinostat (1 µM or 10 µM) to investigate its anti-inflammatory effect. After 6 h of incubation, the cells were collected and the expression of IL-1β was evaluated by qPCR [74].

In the experiment, specific treatments were used: (1) control cell culture (no treatment), (2) cell culture stimulated with LPS (0.3 µg/mL), (3) cell culture incubated with 1 µM Givinostat, (4) cell culture incubated with 10 µM Givinostat, (5) cell culture stimulated with LPS (0.3 µg/mL) and Givinostat (1 µM) and (6) cell culture stimulated with LPS (0.3 µg/mL) and Givinostat (10 µM).

### 4.2. Experimental Neonatal Hypoxia-Ischemia (In Vivo Experiments)

All experiments were conducted with an approved protocol from the 2nd Local Ethics Committee for Animal Experimentation in Warsaw, Poland (permit no. WAW2/081/2018) according to EU Directive 2010/63/EU. All experiments and methods were performed in accordance with the relevant regulations and ARRIVE guidelines.

Neonatal hypoxia-ischemia (HI) was induced in postnatal day 7 (PND7) Wistar rats of both sexes based on the method developed by Rice et al. [75]. Briefly, the rat pups were anesthetized with isoflurane (4% induction, 2% maintenance) and then the left common carotid artery was double ligated with surgical silk and cut between ligatures. The incision was sutured and the wound was treated with lignocaine as an analgesic. The entire surgical procedure lasted for a maximum of 5 min to avoid the neuroprotective influence of isoflurane. Then, the rat pups were returned to their home cage for 1 h recovery. To induce hypoxia, the operated animals were placed for 1 h in a chamber containing 7.6% oxygen in nitrogen at a controlled temperature (35 °C). Sham-operated animals served as controls and underwent the same surgical procedure without ligation of the carotid artery and without hypoxia. After the entire procedure, the pups were returned to their home cage, housed at 22 ± 2 °C and 55 ± 5% humidity following a 12/12 h light/dark cycle and were fed by their dams. In this HI model, a hypoxic-ischemic lesion occurred in the cerebral hemisphere on the side of the ligated artery (ipsilateral), and only these hemispheres were used in the experiments.

Rats from each litter were randomly assigned to 4 experimental groups (5 rats per group and time point): (1) control sham-operated animals (Ctr), (2) control sham-operated animals treated with Givinostat (Ctr+Giv), (3) animals after hypoxia-ischemia (HI) and (4) animals after hypoxia-ischemia treated with Givinostat (HI+Giv).

### 4.3. Drug Treatment

Control and HI rats were subcutaneously injected with Givinostat (Sigma–Aldrich, Burlington, MA, USA) (10 mg/kg body weight) [17] or a vehicle (10% DMSO in saline) at the same volume starting immediately after hypoxic exposure and lasting 5 consecutive days (one dose every 24 h).

### 4.4. Tissue Preparation

Rats surviving 7 or 14 days after HI or aged-matched controls were deeply anesthetized with intraperitoneal injections of 100 mg/kg body weight ketamine combined with 10 mg/kg body weight xylazine and were perfused transcardially first with phosphate-buffered saline (PBS) and then with a fixative solution of 4% paraformaldehyde in PBS (pH 7.4). Afterwards, the brains were dissected, postfixed for 3 h at 4 °C in the same fixative solution, cryoprotected overnight in a 30% sucrose solution and finally frozen on dry ice. Coronal cryostat brain sections (30 µm thick) were cut (from Bregma −2.80 mm Lambda 4.80 mm to Bregma 3.40 mm Lambda 4.20 mm) in serial order to create 10 series sections and were used for immunohistochemical studies.

Western blot and Luminex analyses were performed on nonperfused brains. Rats 24 h, 72 h and 5 days after HI and aged-matched controls were deeply anesthetized with ketamine and xylazine (as described above) and decapitated. From the skull of the HI animals, ipsilateral hemispheres (injured/hypoxic-ischemic) were dissected. In the case of the control animals, only the left hemispheres were analyzed. Then, the brain hemispheres were homogenized in an RIPA lysis buffer (10 mM Tris-HCl pH 7.5 containing 150 mM NaCl, 1% Nonidet P40, 0.1% SDS, 1% Triton X-100, PMSF 0.1 mg/mL) supplemented with proteinase and a phosphatase inhibitor cocktail (1:100, Sigma Aldrich cat. no. PPC1010, St. Louis, MO, USA). The lysates were centrifuged at 13,000× *g* for 10 min at 4 °C. The supernatant was collected and used to measure cytokine and chemokine expression (by Luminex method) and CX3CL1 protein levels (by Western blot method). Cell pellets were resuspended in the RIPA lysis buffer and were used for acetyl-H3 analysis. The total protein concentrations (in the supernatant as well as in the pellet solution) were assessed by a Bio-Rad DCTM protein assay kit (Bio-Rad, Hercules, CA, USA). Then, the lysates were frozen on dry ice and were stored at −80 °C until use.

### 4.5. Quantitative Polymerase Chain Reaction (qPCR)

IL-1β expression was assessed in glial cells stimulated with LPS and Givinostat. The total RNA was isolated with a Total RNA Mini Kit (A&A Biotechnology, Gdańsk, Poland) according to the manufacturer’s instructions, and the quality and concentration of the RNA were verified by spectrophotometry with a NanodropTM apparatus (Thermo Fisher Scientific, Waltham, MA, USA). The samples containing 250 ng of the total RNA were reverse transcribed using a High Capacity RNA-to-cDNA Kit (Thermo Fisher Scientific) according to the manufacturer’s instructions.

Quantitative real-time PCR analyses of the cDNA samples (2.5 ng) with a Fast SYBR Green Master Mix (Thermo Fisher Scientific) and primers specific for IL-1β (forwards—CACCTCTCAAGCAGAGCACAG, reverse—GGGTTGCATGGTGAAGTCAAC) and β-actin (forwards—CGGTCAGGTCATCACTATCG, reverse—TTCCATACCCAGGAAGGAAG) were performed in a 7500 Fast Real-Time PCR System (Applied Biosystem). The reaction parameters were as follows: (1) holding stage, 10 s at 95 °C; (2) cycling stage (40×), 3 s at 95 °C, 30 s at 60 °C and 45 s at 72 °C; and (3) melt curve stage, 15 s at 95 °C, 1 min at 60 °C, repeated in two cycles. Each sample was tested in triplicate during two analysis sessions. The fluorescence signal from a specific transcript was normalized against that of the reference gene (β-actin), and the threshold cycle values (ΔCt) were quantified as fold changes by the 2^−ΔΔCT^ method.

### 4.6. Immunohistochemical Staining

The following antibodies were used (source, catalogue number, final dilution): mouse monoclonal anti-ED1 (CD68) (AbD Serotec, Oxford, UK, MCA341R, 1:100), goat polyclonal anti-Arg-1 (arginase-1) (Santa Cruz, Dallas, TX, USA sc-18354, 1:250), rabbit polyclonal anti-IL-1β (Santa Cruz, Dallas, TX, USA, sc-7884, 1:250), anti-Iba (Abcam, Cambridge, UK, ab5076, 1:200), anti-CX3CR1 (Abcam, Cambridge, UK, ab 8021, 1:100) and anti-CX3CL1 (Invitrogen, Waltham, MA, USA, PA5-47728, 1:100). 

Microglial cells were labeled with an anti-Iba1 antibody specific for all states of microglia (ramified, activated and phagocytic) or anti-ED1 antibody-labeled activated macrophage-like microglia. For the identification of the type of microglia polarization, we used anti-IL-1β for the identification of proinflammatory M1 microglial cells (ED1/ IL-1β double staining) and anti-arginase-1 for the identification of anti-inflammatory M2 microglial cells (ED1/Arg-1 double staining). The expression of CX3CR1 on microglia was evaluated by Iba1/CX3CR1. Double labeling was also employed for monitoring the CX3CR1/CX3CL1 connections. Double fluorescent immunohistochemistry was performed on 30 µm thick free-floating coronal cryostat sections. Adjacent series of brain sections were rinsed in PSB and were incubated in a blocking medium (10% normal goat serum in PBS containing 0.25% Triton X-100) for 1 h at room temperature. Afterwards, sections were incubated overnight at 4 °C with an antibody specific for microglia (Iba1 or ED1) or fractalkine (CX3CL1). Then, tissue sections were rinsed in PBS, and the primary antibodies were revealed by applying appropriate secondary Cy3-conjugated antibodies (AlexaFluor 546, 1:500) for 1 h at room temperature and in the dark. In the next step of double immunofluorescent staining, the sections were rinsed in PBS and were incubated with primary antibodies (anti-Arg-1, anti-IL-1β or anti-CX3CR1) overnight at 4 °C. Then, after rinsing in PBS, the sections were exposed to appropriate FITC-conjugated secondary antibodies (AlexaFluor 488, 1:500) for 1 h at room temperature. The nuclei were subsequently labeled with the fluorescent dye DAPI (300 nM in PBS; Thermo Fisher Scientific).

Labeling was verified using a confocal laser scanning microscope (LSM 780, Carl Zeiss, Jena, Germany) with ZEN software. A helium–neon laser (543 nm) was utilized in the excitation of Alexa Fluor 546, while an argon laser (488 nm) was applied in the excitation of FITC. The number of double-stained cells was manually counted in an average of five brain sections per animal in a 1.44 mm^2^ area of the brain cortex utilizing ImageJ 1.46 software (Rasband, W.S., U. S. National Institutes of Health, Bethesda, MD, USA).

### 4.7. Evaluation of Cytokine and Chemokine Expression after Hypoxia-Ischemia (Luminex)

Concentrations of cytokines and chemokines were analyzed in brain lysates obtained from the control and HI rats untreated or treated with Givinostat by the Bio-Plex Pro Rat Cytokine 23-plex Assay (BioRad, #12005641) method according to the manufacturer’s instructions. The following cytokines and chemokines were measured: interleukin-1α (IL-1α), interleukin-1β (IL-1β), interleukin-2 (IL-2), interleukin-4 (IL-4), interleukin-5 (IL-5), interleukin-6 (IL-6), interleukin-7 (IL-7), interleukin-10 (IL-10), interleukin-12 (IL-12, p70), interleukin-13 (IL-13), interleukin-17A (IL-17A), interleukin 18 (IL-18), interferon-γ (IFN-γ), tumor necrosis factor-α (TNF-α), chemokine C-X-C motif ligand-1 (CXCL-1, GRO/KC), granulocyte colony-stimulating factor (G-CSF), macrophage colony-stimulating factor (M-CSF), granulocyte-macrophage colony-stimulating factor (GM-CSF), C-C motif chemokine ligand 5 (CCL-5, RANTES), monocyte chemoattractant protein-1 (MCP-1), macrophage inflammatory protein-1 α (MIP-1α), macrophage inflammatory protein-3 α (MIP-3α) and vascular endothelial growth factor (VEGF). The median fluorescence intensity plates were assayed on a Bio-Plex^®^200 Luminex system with Bio-Plex Manager 5.0 software. The five-parameter logistic method was applied to estimate cytokine concentrations in brain lysates. Five animals per experimental group were analyzed.

### 4.8. Western Blot Analysis

The levels of histone-3 (H3) acetylation and CX3CL1 in the rat brains after HI and Givinostat treatment were evaluated in tissue pellets or lysates prepared as described above. Samples containing 50 µg of protein were separated by SDS–PAGE and were transferred onto nitrocellulose (AmershamTM ProtranTM Supported 0.45 μm NC). After blocking in 5% nonfat milk, the membranes were incubated at 4 °C overnight with a rabbit polyclonal anti-acetyl-H3 primary antibody (Millipore, Burlington, MA, USA, cat. no 06-599, 1:1000) or rabbit polyclonal anti-CX3CL1 (Invitrogen, cat. no PA1-29025, 1:1000), rinsed 3 times in PBS and then incubated for 1 h at room temperature with an anti-rabbit horseradish peroxidase-conjugated secondary antibody (Sigma-Aldrich). To verify an equal loading of protein per line, a beta-actin antibody (MP Biomedicals, Irvine, CA, USA, cat. no 0869100 MP, 1:500) was used as an internal control for each WB analysis. Immunoblot signals were visualized using an ECL chemiluminescence kit (GE Healthcare Life Sciences, Chicago, IL, USA, cat. no RPN2106) visualized by membrane exposure to an X-ray HyperfilmTM ECL film (GE Healthcare Life Sciences, cat. no 70487). A semiquantitative estimation of acetyl H3 and CX3CL1 levels detected by immunoblotting was performed utilizing LKB Utrascan XL Program GelScan software. The densitometry values were averaged in all groups and then the densitometry values in the control groups were taken as 100%. The data from the respective experimental groups are presented as percentages of the control value.

### 4.9. Statistical Analysis

GraphPad PRISM 5.0 software (Motulsky H., San Diego, CA, USA) was used for the statistical analysis of the data. Comparisons between the animal groups were performed using a one-way analysis of variance (ANOVA) followed by Tukey’s post-hoc test for multiple comparisons. All values are expressed as the mean ± SD. The data were considered significant at a *p* value < 0.05. 

## 5. Conclusions

Our results clearly showed that neonatal hypoxia-ischemia produces an inflammatory response by activating the proinflammatory M1 phenotype of microglia, disrupting the microglia–neuron (CX3CL1/CX3CR1) axis and elevating numerous proinflammatory cytokines/chemokines. An inhibitor of histone deacetylases, Givi-nostat/ITF2357 used in our study as a potential immunosuppressive agent, did not modulate the microglial response after HI and did not suppress most of the molecular inflammatory mediators (chemokines/cytokines) that are crucial for inducing cerebral damage after hypoxia-ischemia. The only anti-inflammatory effect of Givinostat found in this study may be associated with the decrease in MIP-1α. However, as cytokines and chemokines act as a final common pathway to injury, they might be potential targets for therapeutic interventions.

## Figures and Tables

**Figure 1 ijms-23-08287-f001:**
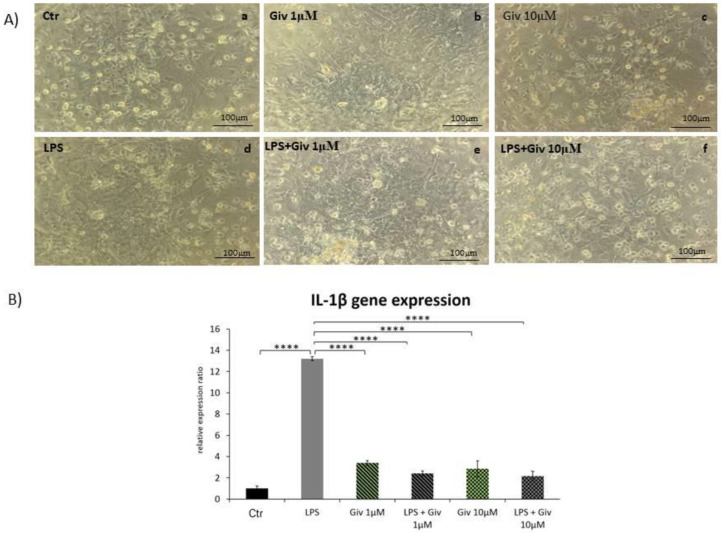
The effect of Givinostat on primary glial cells exposed to LPS. (**A**) Supravital photomicrographs of primary glial cell cultures: untreated (**a**), treated with 1 µM Givinostat (**b**), 10 µM Givinostat (**c**), LPS (0,3 μg/mL) (**d**) and exposed simultaneously to LPS and 1 µM (**e**) or 10 µM Givinostat (**f**). (**B**) IL-1β gene expression level in primary glial cell culture exposed to LPS (0.3 μg/mL) and Givinostat (1 µM or 10 µM). After 6 h of simultaneous incubation with the LPS and HDAC inhibitor, the cells were harvested, and IL-1β gene expression was estimated by qPCR. Note that LPS increased the level of IL-1β-expressing glial cells, whereas the presence of Givinostat restored the expression of this proinflammatory cytokine to the control level. One-way ANOVA indicated significant differences in the level of IL-1β gene expression: **** *p* < 0.0001.

**Figure 2 ijms-23-08287-f002:**
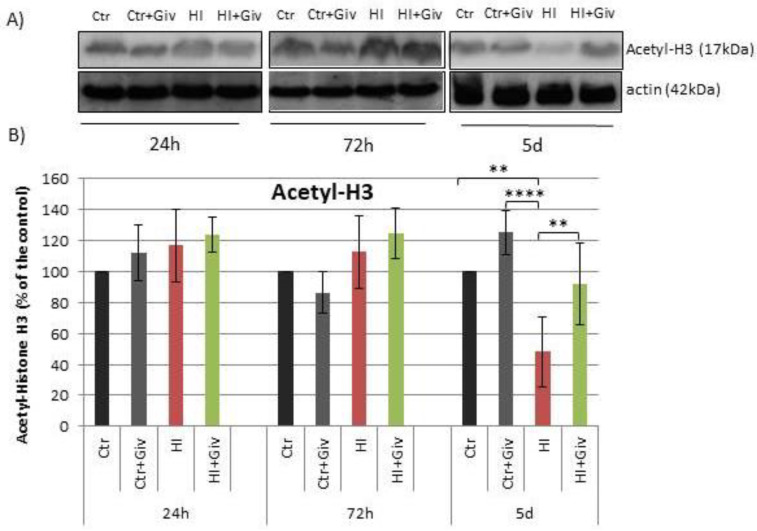
The effect of Givinostat on the acetylation of histone H3 after neonatal hypoxia-ischemia. (**A**) Representative immunoblots of acetylated H3 analyzed 24 h, 72 h and 5 days after HI and in control animals. (**B**) The graph shows the statistical analysis of densitometric data presented as a percent of the control value. The intensity of each band was quantified and normalized in relation to the reference protein (actin). The values are the mean ± SD from five animals per group and time point. Note the decreased level of acetyl-H3 in the hypoxic-ischemic experimental group 5 days after insult. The administration of Givinostat restored the acetylation of H3 to the control level. One-way ANOVA indicated significant differences between the control and HI groups as well as the HI and HI+Giv experimental groups (** *p* < 0.01) and the Ctr +Giv vs. HI group (**** *p* < 0.0001).

**Figure 3 ijms-23-08287-f003:**
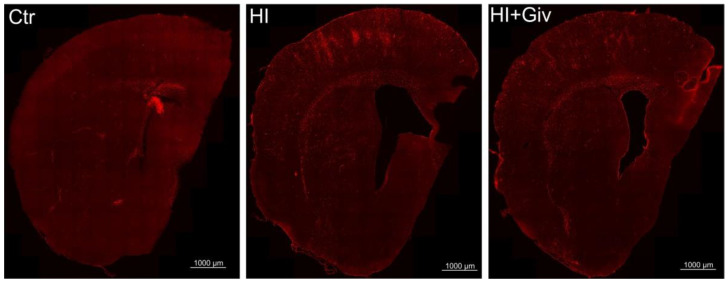
The effect of hypoxia-ischemia and Givinostat treatment on microglia activation pattern in the ipsilateral hemisphere. Confocal photomicrographs of brain sections from animals 7 days after HI, with or without Givinostat application, stained for activated microglia/macrophage marker—ED1 (red). Scale bar: 1000 µm. Confocal photomicrographs of ipsilateral hemispheres show a large number of ED1 stained cells after HI in cortex, striatum and corpus callosum. Application of Givinostat did not change the activation pattern of microglia after HI.

**Figure 4 ijms-23-08287-f004:**
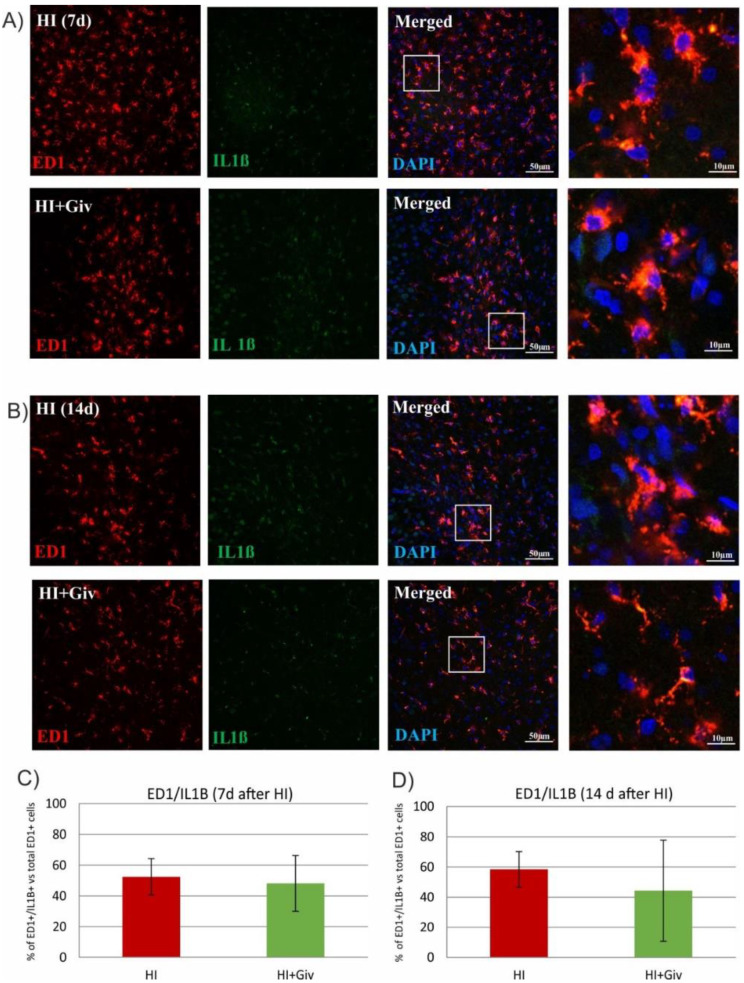
Givinostat has no effect on the polarization of microglia after neonatal hypoxia-ischemia. Sections from ipsilateral (hypoxic-ischemic) hemispheres 7 and 14 days after HI were stained for anti-ED1 antibody-labeled reactive phagocytic microglia (red) and for interleukin-1β (IL-1β), a marker specific for the proinflammatory M1 phenotype (green). Nuclei were labeled with DAPI (blue). Representative confocal photomicrographs of brain sections show double-labeled cells (yellow) 7 days (**A**) and 14 days (**B**) after HI. Enlargements present areas marked in rectangles. Number of ED1/IL-1β-positive cells quantified in the brain cortex area (0.36 mm^2^) 7 days (**C**) and 14 days (**D**) after HI. The values are means ± SD from five animals per group. One-way ANOVA and Tukey’s test did not indicate any significant differences between the investigated groups.

**Figure 5 ijms-23-08287-f005:**
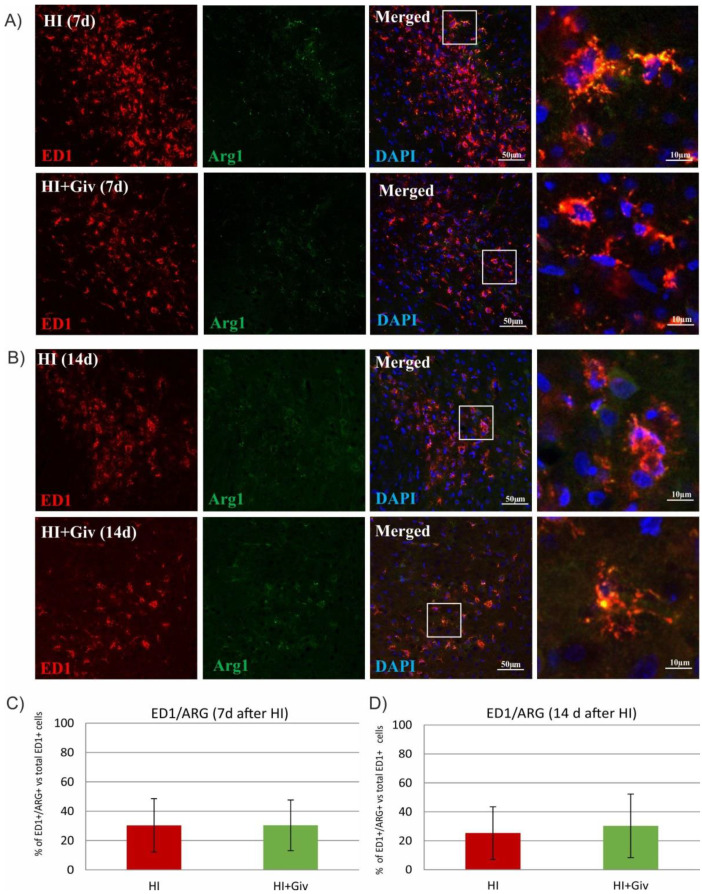
Givinostat has no effect on the polarization of microglia after neonatal hypoxia-ischemia. Sections from ipsilateral (hypoxic-ischemic) hemispheres 7 and 14 days after HI were stained for anti-ED1 antibody-labeled reactive phagocytic microglia (red) and arginase-1 (Arg-1), a marker specific for the M2 phenotype (green). Nuclei were labeled with DAPI (blue). Representative photomicrographs of brain sections show double-labeled cells (yellow) 7 days (**A**) and 14 days (**B**) after HI. Enlargements present areas marked in rectangles. Number of ED1/Arg-1-positive cells quantified in the brain cortex area (0.36 mm^2^) 7 days (**C**) and 14 days (**D**) after HI. The values are means ± SD from five animals per group. One-way ANOVA and Tukey’s test did not indicate any significant differences between the investigated groups.

**Figure 6 ijms-23-08287-f006:**
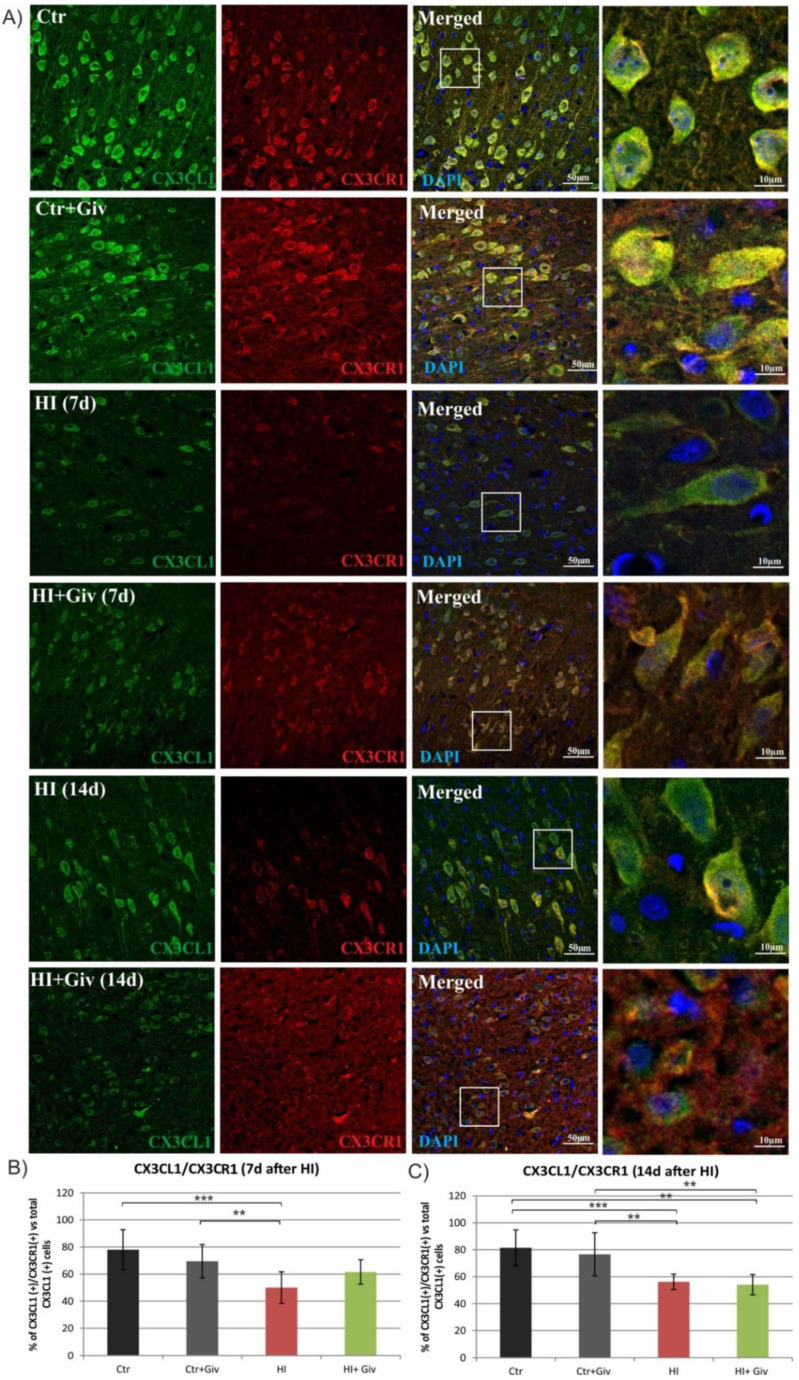
Hypoxia-ischemia disturbed the CX3CL1/CX3CR1 interactions. Sections from ipsilateral (hypoxic-ischemic) hemispheres 7 and 14 days after HI were stained for CXC3L1 antibody (green) and for its receptor CX3CR1 (red) (**B**). Nuclei were labeled with DAPI (blue). (**A**) Representative photomicrographs show colocalization of CX3CL1/CX3CR1 (yellow) in the cortex. Enlargements present areas marked in rectangles. Number of CX3CL1/CX3CR1-positive cells quantified in the brain cortex area (0.36 mm^2^) 7 days (**B**) and 14 days (C) after HI. Note the significant decrease in double-stained cells 7 and 14 days after HI regardless of Givinostat treatment. The values are means ± SD from five animals per group. One-way ANOVA indicates significant differences between the investigated experimental groups: ** *p* < 0.01; *** *p* < 0.001.

**Figure 7 ijms-23-08287-f007:**
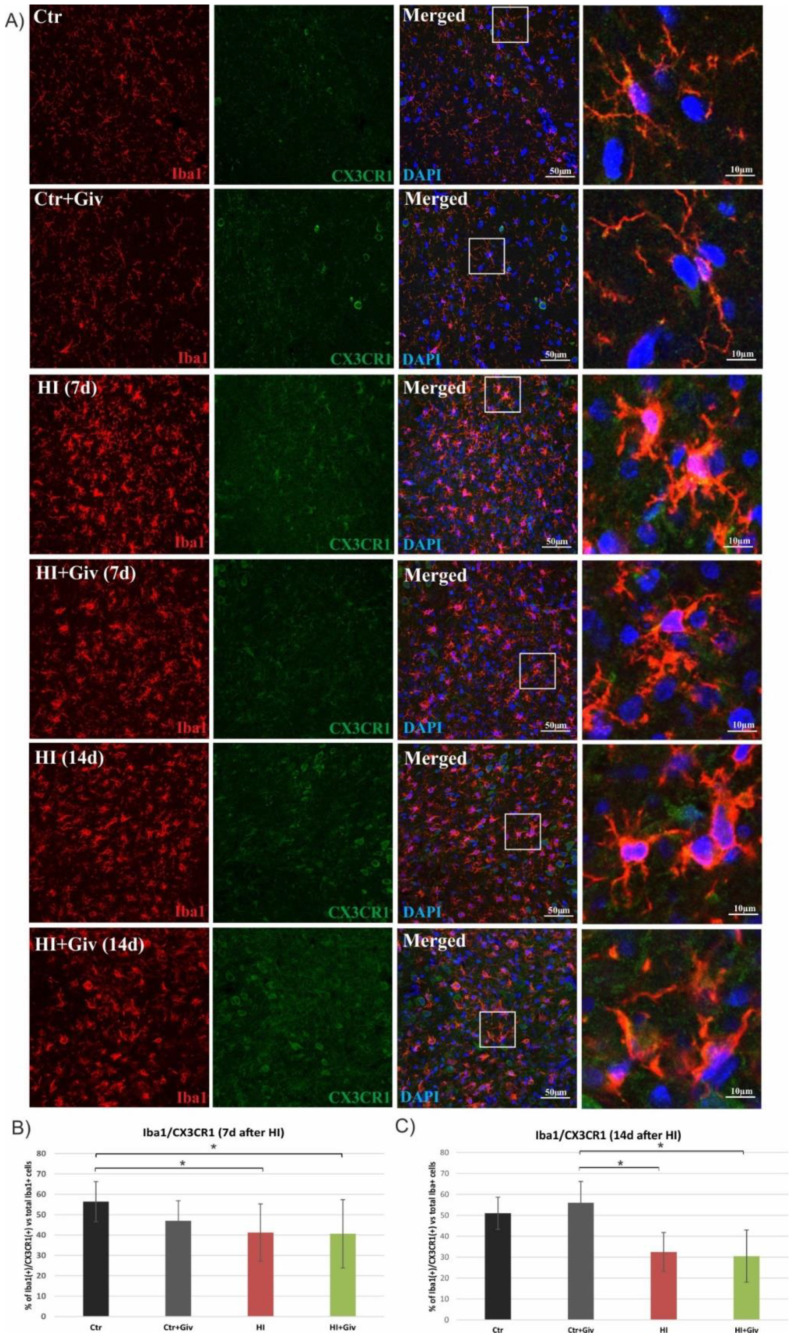
Hypoxia-ischemia decreased the level of CX3CR1 localized on microglia. Sections from ipsilateral (hypoxic-ischemic) hemispheres 7 and 14 days after HI were stained for anti-Iba1 antibody-labeled microglia (red) and for CX3CR1 (green) (**B**). Nuclei were labeled with DAPI (blue). (**A**) Representative photomicrographs show double-labeled cells (yellow) in the cortex. Enlargements present areas marked in rectangles. Number of Iba1/CX3CR1-positive cells quantified in the brain cortex area (0.36 mm^2^) 7 days (**B**) and 14 days (**C**) after HI. Note the significant decrease in CX3CR1 on microglia at both investigated time points after HI regardless of Givinostat administration. The values are means ± SD from five animals per group. One-way ANOVA indicates significant differences between the investigated experimental groups: * *p* < 0.05.

**Figure 8 ijms-23-08287-f008:**
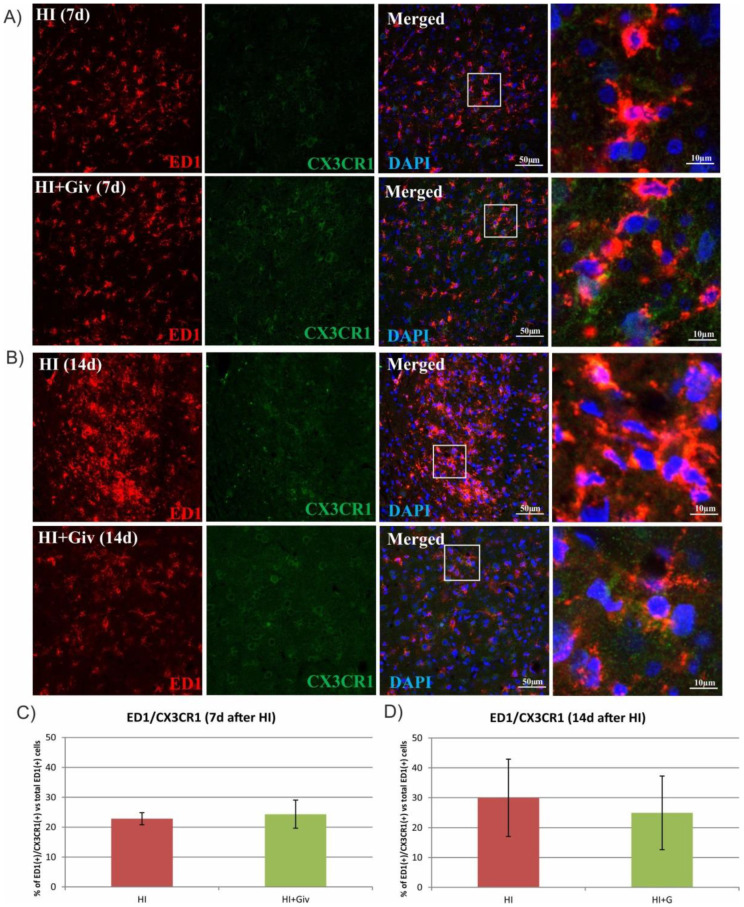
The effect of hypoxia-ischemia on the level of CX3CR1 localized on activated microglia. Sections from ipsilateral (hypoxic-ischemic) hemispheres 7 and 14 days after HI were stained for anti-ED1 antibody-labeled activated microglia (red) and for CX3CR1 (green) (**B**). Nuclei were labeled with DAPI (blue). Representative photomicrographs show double-labeled cells (yellow) in the cortex 7 days (**A**) and 14 days (**B**) after HI. Enlargements present areas marked in rectangles. Number of ED1/CX3CR1-positive cells quantified in the brain cortex area (0.36 mm^2^) 7 days (**C**) and 14 days (**D**) after HI. The values are means ± SD from five animals per group. One-way ANOVA and Tukey’s test did not indicate any significant differences between the investigated groups.

**Figure 9 ijms-23-08287-f009:**
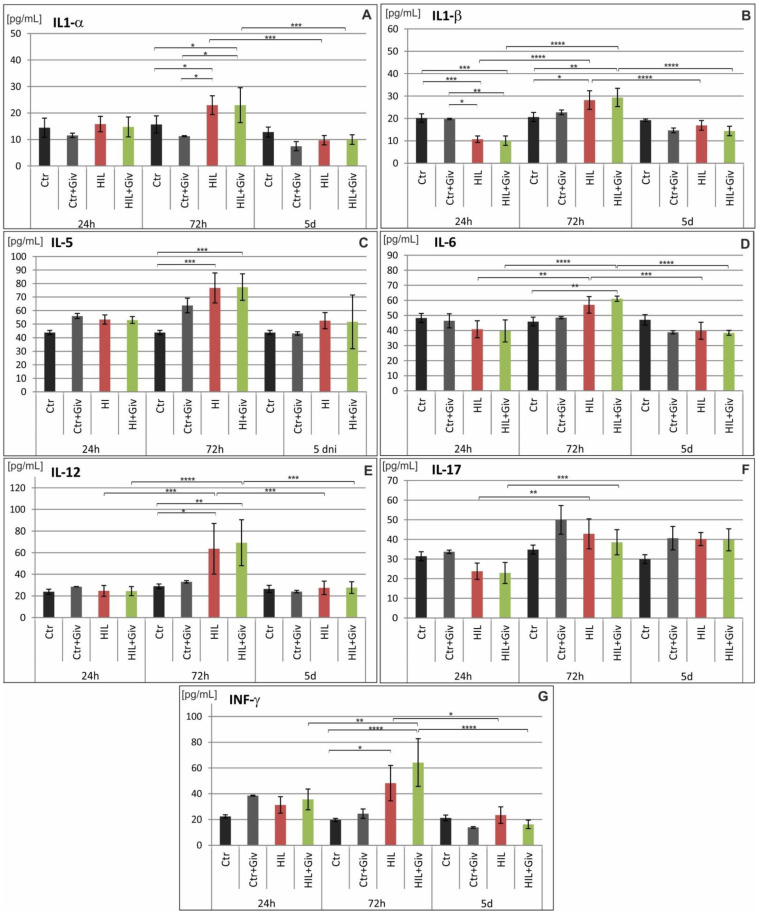
Proinflammatory cytokine profile in neonatal rat brains after hypoxia-ischemia. Quantitation of IL-1α (**A**), IL-1β (**B**), IL-5 (**C**), IL-6 (**D**), IL-12 (**E**), IL-17 (**F**) and IFN-γ (**G**) protein levels detected in the brains of control animals and 24 h, 72 h and 5 days after hypoxia-ischemia with or without Givinostat treatment. Note the elevated level of presented cytokines 72 h after HI regardless of Givinostat administration. The values are the mean ± SD from five animals per group and time point. One-way ANOVA indicates significant differences between the investigated experimental groups: **p* < 0.05; ** *p* < 0.01; *** *p* < 0.001; **** *p* < 0.0001.

**Figure 10 ijms-23-08287-f010:**
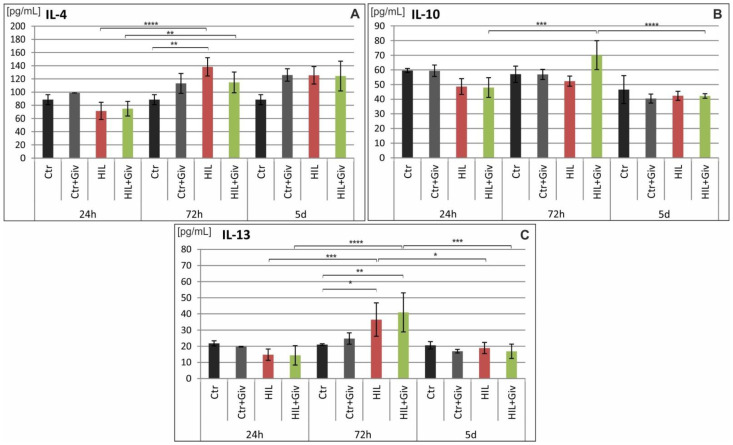
Anti-inflammatory cytokine profile in neonatal rat brains after hypoxia-ischemia. Quantitation of IL-4 (**A**), IL-10 (**B**) and IL-13 (**C**) protein levels detected in the rat brain of control animals and 24 h, 72 h and 5 days after hypoxia-ischemia, treated and untreated with Givinostat. The values are the mean ± SD from five animals per group and time point. One-way ANOVA indicates significant differences between the investigated experimental groups: * *p* < 0.05; ** *p* < 0.01; *** *p* <0.001; **** *p* < 0.0001.

**Figure 11 ijms-23-08287-f011:**
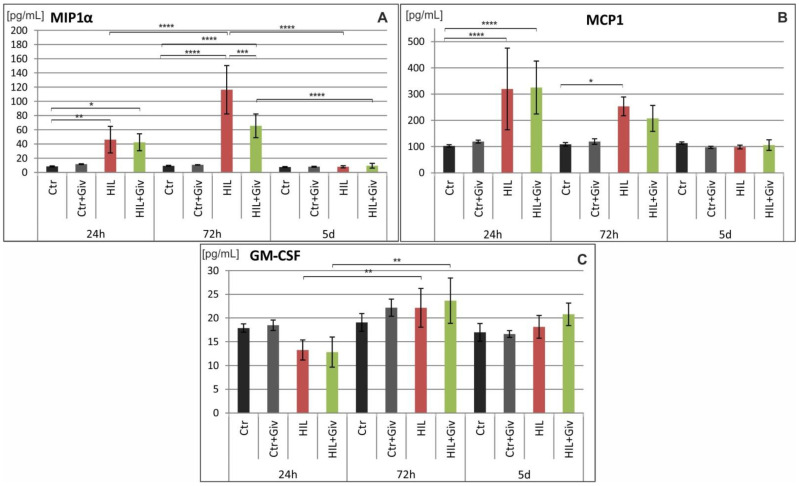
Chemokine profile in neonatal rat brains after hypoxia-ischemia. Quantitation of MIPα1 (**A**), MCP1 (**B**) and GM-CSF (**C**) protein levels detected in the rat brain of control animals and 24 h, 72 h and 5 days after hypoxia-ischemia, treated and untreated with Givinostat. Note the increased levels of MIP-1α and MCP1 24 and 72 h after HI. Givinostat significantly decreased the level of MIP-1α at 72 h after insult. The values are the mean ± SD from five animals per group and time point. One-way ANOVA indicated significant differences between the investigated experimental groups: * *p* < 0.05; ** *p* < 0.01; *** *p* < 0.001; **** *p* < 0.0001.

**Figure 12 ijms-23-08287-f012:**
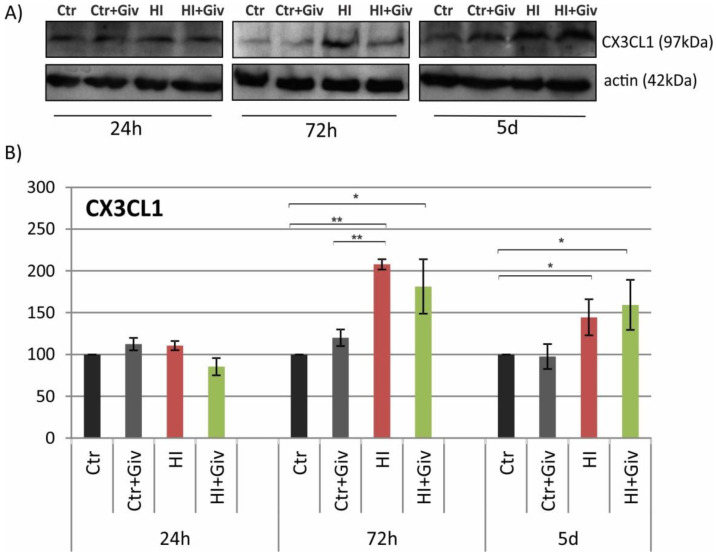
Hypoxia-ischemia increased the level of CX3CL1 in neonatal rat brains. (**A**) Representative immunoblots of CX3CL1 analyzed 24 h, 72 h and 5 days after HI and in age-matched controls. (**B**) The graph shows the statistical analysis of densitometric data presented as a percent of the control value. The intensity of each band was quantified and normalized in relation to the reference protein (actin). The values are the mean ± SD from five animals per group and time point. Note the increased level of CX3CL1 in the hypoxic-ischemic experimental group 72 h and 5 days after insult. The administration of Givinostat after HI did not influence the level of CX3CL1. One-way ANOVA indicates significant differences between the control and HI as well as the control and HI+Giv experimental groups (* *p* < 0.05; ** *p* <0.01).

## Data Availability

The data sets used and/or analyzed during the current study are available from the corresponding author upon reasonable request.

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
