# Peer review of "Analysis of Givinostat/ITF2357 Treatment in a Rat Model of Neonatal Hypoxic-Ischemic Brain Damage"

_ijms, 2022, doi:10.3390/ijms23158287_

Round 1

Reviewer 1 Report

The manuscript by Pawelec et al., on the study of the effects of histone deacetylase inhibitor Givinostat in a rat model of hypoxic-ischaemic brain damage is well written and of interest.

It is unexpected, nonetheless, that the authors were able to identify IL-1beta expression in primary glial cell cultures, which were stimulated by LPS alone. It is generally established that to see an IL-1beta increase in glial cells, a second stimulus, such as ATP, must be given to the cells in addition to the LPS stimulus (see inflammasome mode of action). Could the Authors explain this research protocol?

In the “blot/gels supplemental file” the authors report the original blots used to create figures 2A and 12A. The mean and standard deviation (SD) from five animals per group and time point are the values provided in the densitometric analysis. Is the blot gel performed using a sample from a single animal, or were samples collected and then pulled? Could the Authors provide all the gel blots obtained in their experimental research?

Author Response

First of all, we would like to thank the Reviewers very much for the detailed revision of our manuscript. We really appreciate their time and effort in reviewing this publication.  According to their advice, we made some corrections and improvements to the manuscript. We hope that the Reviewers will accept the present version of our manuscript.

  1. It is unexpected, nonetheless, that the authors were able to identify IL-1beta expression in primary glial cell cultures, which were stimulated by LPS alone. It is generally established that to see an IL-1beta increase in glial cells, a second stimulus, such as ATP, must be given to the cells in addition to the LPS stimulus (see inflammasome mode of action). Could the Authors explain this research protocol?

Answer:  In our in vitro experiment we did not use double stimulation of mixed glial cells with LPS and ATP.  After incubation of these cells only with  LPS at a concentration of 0,3 µg/ml for 6 hours, we noticed 10-fold increase of IL-1beta mRNA expression. We have designed this experiment based on the article:  Faraco et al. : “Histone deacetylase (HDAC) inhibitors reduce the glial inflammatory response in vitro and in vivo.” Neurobiology of Disease; 2009: 36 (2), 269-279. We included this reference in the present version of our manuscript. Moreover, there are other published articles, with the similar experimental protocol  (stimulation of glial cells only with LPS) e.g.:

  • General Anesthetics Inhibit LPS-Induced IL-1β Expression in Glial Cells. Tanaka et al. PloS One; 2013 Dec 11;8(12):e82930.
  • Imipramine and fluoxetine inhibit LPS-induced activation and affect morphology of microglial cells in the rat glial culture. Obuchowicz E. et al. Pharmacological Reports. 2014 Feb;66(1):34-43.
  • Lipopolysaccharide-Induced Microglial Neuroinflammation: Attenuation by FK866. Yaling Xu et al.; Neurochemical Research volume 46, pages 1291–1304 (2021)
  1. In the “blot/gels supplemental file” the authors report the original blots used to create figures 2A and 12A. The mean and standard deviation (SD) from five animals per group and time point are the values provided in the densitometric analysis. Is the blot gel performed using a sample from a single animal, or were samples collected and then pulled? Could the Authors provide all the gel blots obtained in their experimental research?

Answer: In the “blot/gels supplemental file” we demonstrated only representative, non-cropped blots of Acetyl-H3 and CX3CL1, that were used in Figures 2 and 12. During our experiments, we made 5 independent western blot analyses for one-time point using biological material obtained from all experimental rats (5 animals per group, 4 experimental groups (Ctr, Ctr+Givinostat, HI, HI+Givinostat), 3-time points (24h, 72h, 5d) – altogether  15 blots for acetyl H3 and 15 for CX3CL1). In the previous submission, we did not include all the blots in “blot/gels supplemental file”.  In the present version of this supplementary file, we provided all original, non-cropped blots, obtained during our experiments.

Reviewer 2 Report

The paper of Pawelec et al. is well written and well presented. Although many results show the failure of treatment with Givinostat, I believe that even the publication of negative outcomes can be very useful to the scientific world, providing an extra key to understanding.

Comments

- Lines 44-45: "The application of potential neuroprotective agents has been truly restricted due to insufficiency and / or serious side effects demonstrated by their impact on normal brain activity." please after these sentences it would be useful to provide a brief description of the current therapies in use and their side effects; this would better understand why it is necessary to search for new therapeutic strategies.

- The authors mostly evaluated microglia markers (CD68, Iba1..). Have they considered evaluating astrogliosis markers such as GFAP?

- As indicated in the MDPI Instructions for authors "The abstract should be a single paragraph" without headings.

- Control and HI rats were subcutaneously injected with Givinostat (Sigma–Aldrich) (10 mg/kg body weight). On what basis did the authors choose the dose of Givinostat as well as the route of administration?

-  In the conclusions the authors rightly pointed out the limitations of the study. It should also be argued how these results might differ using another experimental model (eg MCAO). In this regard, testing Givinostat in other experimental models could be one of the authors' future goals.

Author Response

First of all, we would like to thank the Reviewers very much for the detailed revision of our manuscript. We really appreciate their time and effort in reviewing this publication.  According to their advice, we made some corrections and improvements to the manuscript. We hope that the Reviewers will accept the present version of our manuscript.

-  Lines 44-45: "The application of potential neuroprotective agents has been truly restricted due to insufficiency and / or serious side effects demonstrated by their impact on normal brain activity." please after these sentences it would be useful to provide a brief description of the current therapies in use and their side effects; this would better understand why it is necessary to search for new therapeutic strategies.

Answer: We have added a brief description of the current therapies in use and their side effects to the Introduction according to Reviewer suggestion:

The only available effective treatment, therapeutic hypothermia, neither provides complete brain protection nor stimulates the repair necessary for the neurodevelopmental outcome. Despite hypothermia treatment about  50% of  neonates with moderate or severe hypoxic-ischemic experience disability or death, thus therapies that improve outcomes are extremely needed.  In recent years several strategies for treatment of neonatal HIE were tested in clinical trials (e.g. erythropoietin, allopurinol, melatonin, cannabidiol, doxycycline, minocycline, exendin-4/exenatide), however most of them cause side effects ( e.g. arthralgia, embolism and thrombosis, hypertension, influenza-like illness, skin reactions, abnormal behavior, insomnia,  fever,  diarrhoea, vomiting, tremor,  stroke). (Victor et al, 2022). Therefore, there is still an urgent need to identify new compounds that may be hopefully adapted as a therapeutic option in infants with hypoxic-ischemic insult.

- The authors mostly evaluated microglia markers (CD68, Iba1..). Have they considered evaluating astrogliosis markers such as GFAP?

Answer: In the present project we did not evaluate the influence of Givinostat on astrogliosis observed after neonatal hypoxia-ischemia. However, in our previous study we determined the influence of another HDAC inhibitor- sodium butyrate on the astroglial response after neonatal HI.  Sodium butyrate increased the number of reactive astrocytes in the rat ipsilateral hemisphere after HI, however, decreased IL-1β expression in these cells (Jaworska et al, J Neuroinflammation. 2017 Feb 10;14(1):34. doi: 10.1186/s12974-017-0807-8). Sodium butyrate also changed the polarization of microglia after neonatal HI (from M1 to M2), as well as decreased inflammatory markers in the rat brains after HI.  We did not see a similar effect of  Givinostat in the same experimental model. However, we cannot exclude some positive effects of Givinostat on astrogliosis after HI. We will assess this issue in our future experiments.   

- As indicated in the MDPI Instructions for authors "The abstract should be a single paragraph" without headings.

 Answer: We corrected the abstract according to MDPI instructions.

- Control and HI rats were subcutaneously injected with Givinostat (Sigma–Aldrich) (10 mg/kg body weight). On what basis did the authors choose the dose of Givinostat as well as the route of administration?

 Answer: The dosage of Givinostat (10 mg/kg body weight) was selected in accordance with previous studies utilizing Givinostat/ITF2357 : Shein et al. “Histone deacetylase inhibitor ITF2357 is neuroprotective, improves functional recovery, and induces glial apoptosis following experimental traumatic brain injury ( FASEB J. 2009;23(12):4266-4275. doi:10.1096/fj.09-134700).  The authors did not observe any adverse effects or mortality among treated animals. The route of administration of other HDACi- sodium butyrate was determined experimentally in our previous study (Ziemka-Nalecz et al. Mol Neurobiol. 2017 Sep;54(7):5300-5318. doi: 10.1007/s12035-016-0049-2), and in the present study we applied the same scheme of Givinostat administration after experimental HI.  

-  In the conclusions the authors rightly pointed out the limitations of the study. It should also be argued how these results might differ using another experimental model (eg MCAO). In this regard, testing Givinostat in other experimental models could be one of the authors' future goals.

Answer:  We would like to thank the reviewer for the comment, however, we have to kindly disagree with the reviewer’s remark. In our opinion Conclusion section should contain only the general summary of the results presented in the article. The differences between our results and the results of other authors were presented in the Discussion section. To my knowledge there are no publications describing the effect of Givinostat in other animal models of ischemia (e.g MCAO),  therefore we cannot speculate about the effect of this HDAC inhibitor. However, we cannot exclude the neuroprotective or anti-inflammatory effect of Givinostat in different animal models of this disorder. 

Round 2

Reviewer 1 Report

The manuscript is now suitable for publication.

Reviewer 2 Report

To my knowledge, the conclusions section in addition to giving a summary of the work can also contain possible future perspectives and further overviews. However, I respect the authors' opinion.

For me the paper is now acceptable.